# Reward Shifting for Optimistic Exploration and Conservative Exploitation

## Abstract

In this work, we study the simple yet universally applicable case of reward shaping, the linear transformation, in value-based Deep Reinforcement Learning. We show that reward shifting, as the simplest linear reward transformation, is equivalent to changing initialization of the $Q$-function in function approximation. Based on such an equivalence, we bring the key insight that a positive reward shifting leads to conservative exploitation, while a negative reward shifting leads to curiosity-driven exploration. In this case, a conservative exploitation improves offline RL value estimation, and the optimistic value estimation benefits the exploration of online RL. We verify our insight on a range of tasks: (1) In offline RL, the conservative exploitation leads to improved learning performance based on off-the-shelf algorithms; (2) In online continuous control, multiple value functions with different shifting constants can be used to trade-off between exploration and exploitation thus improving learning efficiency; (3) In online RL with discrete action space, a negative reward shifting brings an improvement over the previous curiosity-based exploration method.

## 1 Introduction

While reward shaping is a well-established practice in reinforcement learning applications and has a long-standing history (Randløv & Alstrøm, 1998; Laud, 2004), specifying a certain reward to incentivize the learning agent requires domain knowledge and deep understanding of the task (Vinyals et al., 2019; Akkaya et al., 2019; Berner et al., 2019; Elbarbari et al., 2021). Even with careful design and tuning, learning with a shaped reward that intends to accelerate learning may on the contrary hinder the learning performance by incurring the sub-optimal behaviors of the agent (Florensa et al., 2017; Plappert et al., 2018). Although Ng et al. (1999) theoretically points the optimal policy will keep unchanged under a special form of reward transformation, and in the later work of Wiewiora et al. (2003) a framework is proposed to guide policies with prior knowledge under tabular setting, the investigation of how it accommodates recent Deep Reinforcement Learning (DRL) algorithms remains much less explored.

In this work, we focus on the simplest case of reward shaping, the linear transformation, in value-based DRL (Sutton & Barto, 1998; Lillicrap et al., 2015; Mnih et al., 2015; Fujimoto et al., 2018b). We start with understanding how such a specific kind of reward shaping works in value-based DRL algorithms. We show that reward shifting, as the simplest reward transformation, is equivalent to engineering initialization of the $Q$-function estimation, extending previous discovery of (Wiewiora et al., 2003) to the function approximation settings. Based on such an equivalence, we bring the key insight of this work: a positive reward shifting leads to conservative exploitation, while a negative reward shifting leads to curiosity-driven exploration. We demonstrate the application of such an insight to three downstream tasks: (1) for offline RL, we show that conservative exploitation can lead to improved learning performance based on off-the-shelf algorithms; (2) for online RL setting, we show multiple value functions with different reward shifting constants can be used as a trade-off between exploration and exploitation, thus improving learning efficiency; (3) finally, we introduce a simple yet crucial improvement over a prevailing curiosity-based exploration method, the Random Network Distillation (Burda et al., 2018b), making it compatible with value-based DRL algorithms. We evaluate our idea on various tasks, including both continuous and discrete action space control, resulting in substantial improvements over previous baselines.

Our contributions can be summarized as follows

1. we introduce the key insight that reward shifting is equivalent to diversified $Q$-value network initialization, which can be used to boost both curiosity-driven exploration and conservative exploitation;

2. motivated by our key insight, we present three scenarios where the reward shifting can benefit, namely the offline conservative exploitation, the online sample-efficient RL, and the curiosity-driven exploration;

3. we demonstrate the effectiveness of the proposed method integrated with off-the-shelf baselines on both continuous and discrete control tasks.

## 2 PRELIMINARIES

### 2.1 ONLINE RL

We follow a standard MDP formulation in the online RL settings, i.e., $\mathcal{M} = \{\mathcal{S}, \mathcal{A}, \mathcal{T}, \mathcal{R}, \rho_0, \gamma, T\}$, where $\mathcal{S} \subset \mathbb{R}^d$ denotes the $d$-dim state space, $\mathcal{A}$ is the action space (note for discrete action space $|\mathcal{A}| < \infty$ and for continuous control $|\mathcal{A}| = \infty$), $\mathcal{T} : \mathcal{S} \times \mathcal{A} \mapsto \mathcal{S}$ is the transition dynamics, $\mathcal{R} : \mathcal{S} \times \mathcal{A} \mapsto \mathbb{R}$ is the reward function. $\rho_0$ denotes the initial state distribution, i.e., $\rho_0 = p(s_0)$. $\gamma$ is the discount factor and $T$ is the episodic decision. Online RL considers the problem of learning a policy $\pi \in \Pi : \mathcal{S} \mapsto \Delta\mathcal{A}$ (or $\pi \in \Pi : \mathcal{S} \mapsto \mathcal{A}$ with a deterministic policy class), such that the expected cumulative reward in the Markov decision process is maximized, i.e.,

$$\pi = \arg\max_{\pi} \mathbb{E}_{a_t \sim \pi, s_{t+1} \sim \mathcal{T}, s_0 \sim \rho_0} \sum_{t=0}^{T} \gamma^t r_t(s_t, a_t), \tag{1}$$

In the online RL setting, an agent normally learns through trials and errors (Sutton & Barto, 1998), either with an on-policy paradigm (Schulman et al., 2015; 2017; Cobbe et al., 2021) or an off-policy manner (Mnih et al., 2015; Lillicrap et al., 2015; Wang et al., 2016; Haarnoja et al., 2018; Fujimoto et al., 2018b). In this work, we focus on the off-policy value-based methods which are in general more sample efficient. Specifically, our discussions assume the policy learning is based on a learned $Q$-value function, that approximates the cumulative reward an agent can gain in the following part of an episode. The $Q$-value function is defined as $Q(s_t, a_t) = \mathbb{E}_{\pi, \mathcal{T}} \sum_{\tau=t}^{T} \gamma^t r(s_\tau, a_\tau)$, and can be approximated through the Bellman Operator $\mathbb{B}Q(s, a) = r(s, a) + \gamma \mathbb{E}\bar{Q}(s', a')$. For value-based methods, the (soft-)optimal policy is then produced by

$$\pi_\alpha^*(a|s) = \frac{\exp \frac{1}{\alpha} Q^*(s, a)}{\sum_{a'} \exp \frac{1}{\alpha} Q^*(s, a')}, \tag{2}$$

where $Q^*$ is optimal $Q$-value function. We can also set the temperature parameter close as 0 to have the deterministic policy class. Simplifying the notion we have $\pi(s) = \arg\max_a Q^*(s, a)$. Algorithms like DPG (Silver et al., 2014) can be used to address the intractable analytical argmax issue arises in continuous action space. We develop our work on top of prevailing baseline algorithms of DQN (Mnih et al., 2015), BCQ (Fujimoto et al., 2018a), and TD3 (Fujimoto et al., 2018b), and it will be easy to extend to other baseline algorithms.

### 2.2 EXPLORATION AND THE CURIOSITY-DRIVEN METHODS

One of the most important issues in online RL is the exploration-exploitation dilemma (Sutton & Barto, 1998) that the agent must learn to exploit its accumulated knowledge on the task while exploring new states and actions. Plenty of previous works address the exploration problem from various perspectives: In the tasks with discrete action space, count-based methods like Bellemare et al. (2016); Ostrovski et al. (2017); Tang et al. (2017) are proposed to motivate the policy to explore more on under-explored states. Curiosity-driven methods are investigated by Houthooft et al. (2016); Pathak et al. (2017); Burda et al. (2018a;b), where the intrinsic reward is designed as a supplementary to the primal task reward for better exploration. Self-imitate approaches like Oh et al. (2018); Ecoffet et al. (2019); Sun et al. (2019) repeat success trajectories but require extra

assumptions on the environment. The work of DIAYN and DADS (Eysenbach et al., 2018; Sharma et al., 2019) show that various skills can be developed even without the primal extrinsic reward. For continuous control tasks, OAC (Ciosek et al., 2019) improves the SAC (Haarnoja et al., 2018) with informative action space noise based on the optimism in face of uncertainty (OFU) (Brafman & Tennenholtz, 2002; Jaksch et al., 2010; Azar et al., 2017; Jin et al., 2018). GAC (Tessler et al., 2019) addresses the exploration issue with a richer functional class for the policy.

In the recent work of Rashid et al. (2020), the problematic pessimistic initialization is addressed for better exploration, yet the work focuses on specific settings of tabular and discrete control exploration. In the work of Osband et al. (2016; 2018), ensemble models with diverse initialization and randomized priors are used to resemble the insight of bootstrap sampling and facilitate better value estimation, yet those methods are only applicable to discrete control tasks. Noted that although the reward shifting can be regarded as a special case of these random priors, it can be distinguished by not changing the optimal Q-value, and flexible to be plugged in to both continuous and discrete control algorithms.

The Random Network Distillation (RND) (Burda et al., 2018b) propose to use the difference between a fixed neural network $\phi_1$ and a trainable network $\phi_2$ to represents the intrinsic reward, e.g.,

$$r_{\text{int}}(s, a) = |\phi_2(s, a) - \phi_1(s, a)|, \tag{3}$$

when outputs of both networks are activated by a sigmoid function, and $\phi_2$ is optimized to approximate $\phi_1$ for the visited $(s, a)$ pairs. Henceforth, the value of $r_{\text{int}}(s, a)$ will decay to 0 when such state-action pairs are visited frequently but remain high for seldom visited pairs.

In this work, we show that exploratory behavior can be achieved simply by shifting the reward function with a constant, thus our method is orthogonal to those previous approaches in the sense that our intrinsic exploration behavior is motivated by function approximation error. We demonstrate such an insight by showing that RND, with its original design, is not suitable for value-based methods in developing exploratory behaviors, but integrating RND with a shifted reward function can remarkably improve the learning performance.

### 2.3 OFFLINE RL

The offline RL, also known as batch-RL, focuses on the problems where the interaction with the environment is impossible, and the policy can only be optimized based on the logged dataset. In those tasks, a fixed buffer $\mathcal{B} = \{s_i, a_i, r_i, s_i'\}_{i=[N]}$ is provided. As the agent in the offline RL setting can not correct its potentially biased knowledge through interactions, the most important issue is to address the extrapolation error (Fujimoto et al., 2018a) induced by distributional mismatch (Kumar et al., 2019). To address such an issue, a series of algorithms optimize the policy learning under the constraint of distributional similarity (Kumar et al., 2019; Wu et al., 2019; Siegel et al., 2020).

Bharadhwaj et al. (2020) proposed CQL to solve the offline RL tasks with a conservative value estimation. Specifically, CQL learns the $Q$-value estimation by jointly maximizing the $Q$-values of actions sampled from the behavior offline dataset and minimizing the $Q$-values of actions sampled with pre-defined prior distributions (e.g., uniform distribution over the action space). As we will show in this work, an alternative approach to have a lower bound for the optimal $Q$-value function is to use an appropriately shifted reward function. This idea leads to the direct application of our proposed framework in the offline setting. In general, reward shift can be plugged in many distribution-matching offline-RL algorithms (Fujimoto et al., 2018a; Kumar et al., 2019; Wu et al., 2019; Siegel et al., 2020) to further improve the performance with conservative $Q$-value estimation.

## 3 A MOTIVATING EXAMPLE

We start with a motivating example that may look counterintuitive: in the prevailing continuous control environment of Humanoid, a three-dimensional bipedal robot is simulated based on the MuJoCo engine. The learning objective is to control the robot to walk forward as fast as possible, without falling over. There are two types of positive reward signals: (1) to encourage forward-moving, i.e., the reward is proportional to the horizontal displacement, and

(2) to encourage the robot to avoid falling over, i.e., a binary alive bonus whether the mass center of the robot is higher than some certain threshold. Although the design of (2) is intended to provide a curriculum for the agent, we show in Figure 1 that such a curriculum, on the contrary, hinders the learning performance of such a bipedal robot to run fast.

The orange curve, TD3 w/o Alive Bonus, shows the smoothed learning curve of a TD3 agent trained with the alive bonus set to be 0; the blue curve, TD3 w/ Alive Bonus, shows the averaged learning curve of a vanilla TD3 agent trained with the alive bonus set to be 5 as default. Noted that for both approaches, we use identically the same primal task in policy evaluation to make sure the curves are comparable. In general curriculum learning improves learning efficiency (Graves et al., 2017; Matiisen et al., 2019; Portelas et al., 2020), how-ever, in the environment of Humanoid, such a curriculum design, which aims at helping the agent learning to run after knowing how to avoid falling over, hinders the learning efficiency and asymptotic performance.

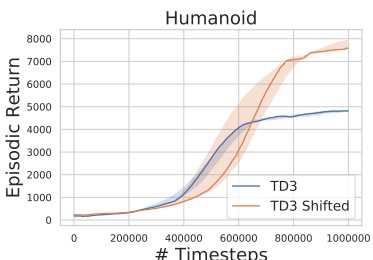

Figure 1: Humanoid agents trained with a linear reward shift drastically gain asymptotic performance.

**Remark 1.** Given an MDP $\mathcal{M} = \{\mathcal{S}, \mathcal{A}, \mathcal{T}, \mathcal{R}, \rho_0, \gamma, T\}$, where $|\mathcal{A}| < \infty$, scaling the reward function with linear transformation, i.e., $\mathcal{R}_{k,b} = k \cdot \mathcal{R} + b, \forall k > 0, b \in \mathbb{R}$, do not change the optimal policy induced by

$$\pi^*(s) = \arg\max_{a \in \mathcal{A}} Q^*_{k,b}(s,a) = \arg\max_{a \in \mathcal{A}} kQ^*(s,a) + \frac{b}{1-\gamma} = \arg\max_{a \in \mathcal{A}} Q^*(s,a), \quad (4)$$

**Remark 2.** Given an MDP $\mathcal{M} = \{\mathcal{S}, \mathcal{A}, \mathcal{T}, \mathcal{R}, \rho_0, \gamma, T\}$, where $|\mathcal{A}| = \infty$, scaling the reward function with linear transformation, i.e., $\mathcal{R}_{k,b} = k \cdot \mathcal{R} + b, \forall k > 0, b \in \mathbb{R}$, do not change the optimal policy induced by deterministic policy gradient (Silver et al., 2014), given proper learning rate:

$$\nabla_\theta J(\mu_\theta) = \mathbb{E}_{s_t}[\nabla_a Q^*(s_t, a_t)|_{a_t = \mu_\theta(s_t)} \nabla_\theta \mu_\theta(s_t)] = \mathbb{E}_{s_t}[\nabla_a Q^*_{k,b}(s_t, a_t)|_{a_t = \mu_\theta(s_t)} \nabla_\theta \mu_\theta(s_t)]/k, \quad (5)$$

In the following, we will focus on the scenarios when $k = 1$ to avoid the trivial (though maybe empirically important) discussions on different learning rates. We reveal the importance of selecting the universal bias term, i.e., $b$ in the reward function, through the lens of initialization priors in function approximation. We further show such a bias term can be utilized not only in the online RL settings to improve learning efficiency but also in the offline RL settings to conduct conservative exploitation with batched data.

## 4 SHIFTED PRIORS FOR Q-VALUE ESTIMATION

### 4.1 REWARD SHIFT EQUALS TO DIFFERENT INITIALIZATION

We start by introducing the central idea of this work: reward shift equals different initialization. Specifically, we illustrate the basic idea with Figure 2. The black curves denote the primal optimal $Q$-value functions (e.g., $Q^*$), and the red curves denote the optimal $Q$-value functions with shifted reward $r' = r + b^+$, where $b^+ > 0$ is a positive constant bias, and we will show a different choice of such a bias leads to a different motivation in $Q$-value estimation. The yellow lines denote $Q$-value estimators, e.g., neural network predictions of the corresponding $Q$-values. **(a)** shift the reward function with a positive bias term $b^+$ will lead to an uniformly increased $Q$-value function, namely $Q^*_{b^+} = Q^* + \frac{b^+}{1-\gamma}$, during learning, a neural network estimator $\tilde{Q}$ initialized with $\tilde{Q}_0 \approx 0$ is optimized to approximate the $Q$-value functions (e.g., through Temporal Difference or Monte Carlo estimation).

**(b)** for any value-based RL algorithm, the value optimization step can be regarded as a function $F$ that minimizes the difference between the estimated $Q$-value function $\tilde{Q}_t$ and the optimal one $Q^*$, given the interaction experience with the environment (e.g., a replay buffer $\mathcal{B}$ for off-policy methods).

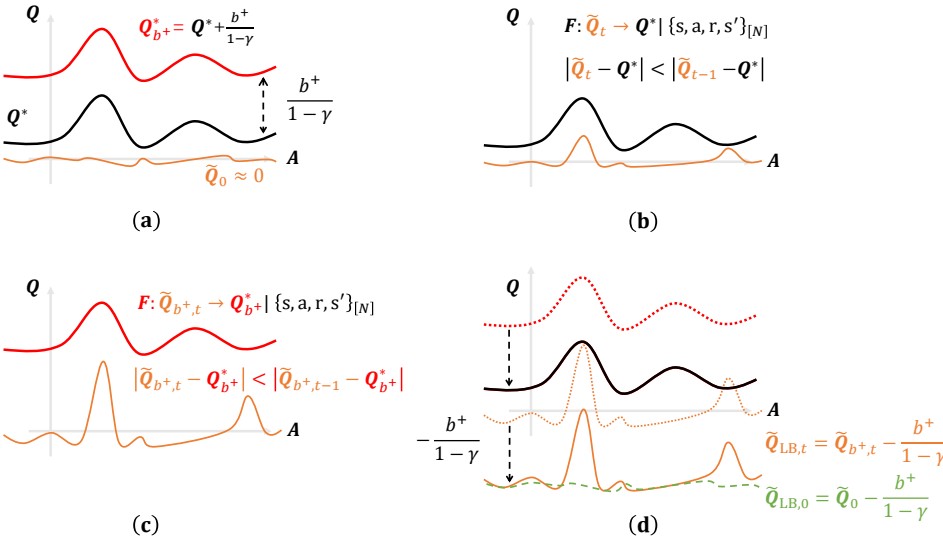

Figure 2: Illustrative figure for conservative exploitation, with a positive constant bias added to the reward function.

(c) similarly, the optimizer given the same interactive experience (e.g., replay buffer $\mathcal{B}$) will learn to minimize the difference between $Q$-value function $\tilde{Q}_{b^+,t}$ and the optimal one $Q^*_{b^+}$, after re-labeling the rewards in the buffer by $r' = r + b^+$.

(d) according to Remark 2, the optimization conducted in (c) is equivalent to (b) with the neural network $Q$-value estimator initialized at $\tilde{Q}_0 \approx 0 - \frac{b^+}{1-\gamma}$, rather than $\tilde{Q}_0 \approx 0$. i.e., by shifting the reward with proper positive value $b^+$, we are able to initialize the $Q$-value network that lower-bounded the optimal $Q$-value.

To summarize, shifting the reward function with a positive constant is equivalent to initializing the value function network with a smaller value, hence during training, the $Q$-value of unseen state-action pair is far lower than the optimal value and hence will not be selected in policy update, leading to conservative learning behavior.

## 4.2 CONSERVATIVE EXPLOITATION

We begin with a natural application in the offline setting, where a policy doesn't interact with the environment and only learns through a logged data set collected from an unknown behavior policy $\pi_\beta$, which can either be an expert that generates high-quality solutions to the task (Fujimoto et al., 2018a; Zhang et al., 2020; Fu et al., 2020) or a non-expert that provides actions that are sub-optimal (Wu et al., 2019; Kumar et al., 2019; Agarwal et al., 2020; Fu et al., 2020; Jarrett et al., 2021) or a mixture of both (Bharadhwaj et al., 2020).

Based on the basic idea we presented in Section 4.1, we introduce a simple yet effective approach for conservative $Q$-value estimation that learns to heuristically put a lower bound on the optimal $Q$-value function in batch settings. Hark back to Figure 2(a), a positive constant $b^+$ added to the reward function will lead to a uniformly (positively) shifted optimal $Q$-value function, and the gap between the primal $Q$-value function and the new one is $\frac{b^+}{1-\gamma}$. Optimizing the $Q$-value function with logged data (e.g., a fixed replay buffer) will minimize the difference between the predicted value and the optimal value with observed data. For the unobserved data point (in the state-action space $\mathcal{S} \times \mathcal{A}$), the near-zero initialization guarantees the prediction is lower than the optimal $Q$-value, thus conservative exploitation can be conducted with such a value function.

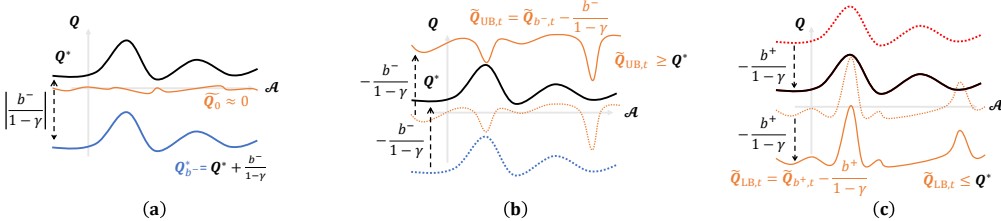

Figure 3: Illustrative figure for curiosity-driven exploration with a negative shifted reward

### 4.3 (Curiosity-Driven) Optimistic Exploration

On the other hand, if we shift the reward function to the negative side, it is equivalent to optimistic initialization. Figure 3 (a-b) illustrate how adding a negative bias leads to curiosity-driven exploration: while adding a negative constant value $b^-$ on the reward function lead to negatively shifted optimal $Q$-value function $Q^*_{b^-}$ (Figure 3 (a)), minimizing the difference between a $Q$-value approximator and the optimal $Q$-value will enable calculating an upper-bound estimation for $Q^*$, as shown in Figure 3(b). With sufficiently large $b^-$ (so that $b^-$ larger than the maximal value of any $s, a$-pair), such an upper bound of $Q^*$ can be used to conduct curiosity-driven exploration. Intuitively, initializing a value network that predicts value larger or equal than the true value will lead to curiosity-driven exploration, as any visited state will be assigned a low value and the policy that learns to perform the action with a higher value will tend to choose novel actions.

Based on the discoveries above that (1) a positive constant shift added to the reward function can be used for conservative policy update, as shown in Figure 3(c) and (2) a negative constant shift added to the reward function can be used for curiosity-driven exploration, as shown in Figure 3(b). We are ready to access both the upper bound, i.e., the optimistic estimation with $b^-$, and the lower bound, i.e., the conservative estimation with $b^+$ of the optimal value function. Henceforth, we are ready to introduce our sample-efficient algorithms for both continuous control and discrete action space respectively. We propose a practical algorithm for general continuous control in Sec. 4.3.1, while focusing on a special class of curiosity-driven exploration method, the RND, in Sec. 4.3.2.

#### 4.3.1 Sample-Efficient Continuous Control with Reward Shift

Based on the principle of optimism on the face of uncertainty (OFU), we introduce a exploration bonus that manifests the uncertainty of the $Q$-value function. Specifically, the basic idea starts from integrating optimistic exploration with conservative exploitation, i.e.,

$$\hat{Q}(s,a) = \tilde{Q}_{\text{LB},t}(s,a) + \beta[\tilde{Q}_{\text{UB},t}(s,a) - \tilde{Q}_{\text{LB},t}(s,a)]$$
$$= (1-\beta)(\tilde{Q}_{b^+,t}(s,a) - \frac{b^+}{1-\gamma}) + \beta(\tilde{Q}_{b^-,t}(s,a) - \frac{b^-}{1-\gamma}) \quad (6)$$
$$= (1-\beta)\tilde{Q}_{b^+,t}(s,a) + \beta\tilde{Q}_{b^-,t}(s,a) - \frac{(1-\beta)b^+ + \beta b^-}{1-\gamma},$$

where the second term with coefficient $\beta$ denotes exploration bonus that is composed of uncertainty.

For those under-explored state-action pairs, i.e., extremely out-of-distribution samples for our neural network, both $\tilde{Q}_{b^+,t}(s,a)$ and $\tilde{Q}_{b^-,t}(s,a)$ will give near-zero predictions as a consequence of initialization (detailed implications are provided in Appendix B). henceforth, the explorative bonus becomes $-\frac{(1-\beta)b^+ + \beta b^-}{1-\gamma}$. Note this is equivalent to applying another constant reward shift with value of $c_r = (1-\beta)b^+ + \beta b^-$.

For the explored state-action pairs close to the samples from the replay buffer, we have

**Proposition 1.** Assuming we have access to an unbiased estimator for the optimal value function $Q^*$, e.g., with Monte-Carlo estimation $\hat{Q}^* = \mathbb{E}\sum_t \gamma r^t$, and the optimization is based on minimizing the MSE between the unbiased estimator and the function approximator, i.e., $\epsilon_t^2 = (\tilde{Q}_t - \hat{Q}^*)^2$,

$\tilde{Q}_t = \tilde{Q}_{t-1} - 2\eta(\tilde{Q}_{t-1} - \hat{Q}^*)$, then combining the linear combination in Equation (6) is equivalent to using a linear combination of the constants with value of $c_r = (1 - \beta)b^+ + \beta b^-$.

According to Proposition 1, a grid search for trading-off between the three hyper-parameters: the exploration bias $b^-$, the exploitation bias $b^+$ and the coefficient in Equation (6) is trivial as they only lead to a linear combination as $c_r = (1 - \beta)b^+ + \beta b^-$. In principle, a meta-learner can be trained to monitor the learning process and select a proper constant automatically (Graves et al., 2017; Matiisen et al., 2019; Portelas et al., 2020). In this work we focus on a simple yet effective uniform sampling strategy from multiple shift constants, which is shown as a strong baseline in Graves et al. (2017); Matiisen et al. (2019), and leave more complicated meta-learner-based approaches in future investigation.

Specifically, we use multiple Q-networks to learn with transition tuples $(s, a, r, s')$ sampled from the identical buffer that collects the policy's interaction history with the environment. For each Q-network, the primal $r$ is replaced with a new reward with shifted constant bias for temporal difference updates. Those learned Q-networks are sampled uniformly during the training of policy network. It is worth noting that our approach only requires post-hoc revision of the primal reward function, rather than interacting with the environment multiple times to collect samples for each value network.

### 4.3.2 IMPROVING VALUE-BASED CURIOSITY-DRIVEN EXPLORATION

The discussion above casts the curiosity-driven exploration method as a special case in the *reward shift equals to initialization* perspective: exploration with RND (Burda et al., 2018b) is equivalent to selecting $b^+ = 0$, i.e., using the primal task reward for exploitation, and using $b^-$ as a fixed random function over $\mathcal{S} \times \mathcal{A}$, i.e., $b^-(s, a) = |\phi_{1,0}(s, a) - \phi_2(s, a)|^2, \forall s, a$, where $\phi_{1,0}$ and $\phi_2$ are two neural networks with different random initializations. While $\phi_{1,0}$ is learnable, $\phi_2$ is set to be fixed during learning as a random curiosity prior (Osband et al., 2018). Without loss of generality, we can assume $\phi_2 = 1$ as a constant initialization, and $\phi_{1,0} = 0$. Then the exploration behavior is fully controlled by the scale of external reward, which is set to be 1.0 with external reward clipped to $[-1, 1]$ in (Burda et al., 2018b). The final equivalent objective in this case is to use $c_r = |\phi_{1,t}(s, a) - 1|^2, t = 0, 1, 2, ...$. Specially, $\phi_{1,0} = 0$ and $c_r = 1$ at beginning.

According to our analysis in the previous section, such a positive shift directly leads to conservative behaviors in $Q$-value estimation, therefore may hinder the exploration behaviors at the beginning, i.e., when $|\phi_{1,t} - 1| \gg 0$. The exploration bonus in RND only becomes effective when visiting seldom visited states after the predictions of $\phi_{1,t}$ of frequently-visited states are close to 1, and hence encouraging stepping into those novel states and discover new knowledge. To overcome the conservative tendency induced by $Q$-value estimation in RND, we propose to use $b^-(s, a) = |\phi_{1,0}(s, a) - \phi_2(s, a)|^2 - I, \forall s, a$, where $I$ is a positive constant (e.g., $I = \max_{s,a} |\phi_{1,0}(s, a) - \phi_2(s, a)|^2$) that assures $b^-(s, a)$ is negative-initialized for optimistic exploratory behaviors.

It is worth noting that the curiosity introduced by exploratory bonuses like RND focuses on increasing the temporary $Q$-value of rarely visited states, which still relies on the visitation behavior itself in random exploration. i.e., without stepping into a non-frequent visited state, the agent will never receive the novelty bonus based on the state. On the other hand, the mechanism of shifting the reward by a constant is an universal optimistic exploration bonus as the curiosity is fused into the initialization across the entire state-action support.

## 5 EXPERIMENTS

### 5.1 OFFLINE REINFORCEMENT LEARNING WITH CONSERVATIVE Q-VALUE ESTIMATION

We first demonstrate the proposed method in the offline RL setting. As has been discussed in Section 4.2, shifting the primal reward function with a positive constant provides a natural way of conservative exploitation. By adding a positive reward shift, any visited $(s, a)$ saved in the given replay buffer $\mathcal{B}$ (logged dataset) will be assigned a large reward, while under-explored $(s, a)$ pairs remain have the low values due to initialization, as illustrated in Figure 2 (d). Although in general our proposed method can be plugged in to any off-the-shelf offline RL algorithm, in this work we demonstrate the effectiveness of such a conservative $Q$-value estimation based on BCQ (Fujimoto

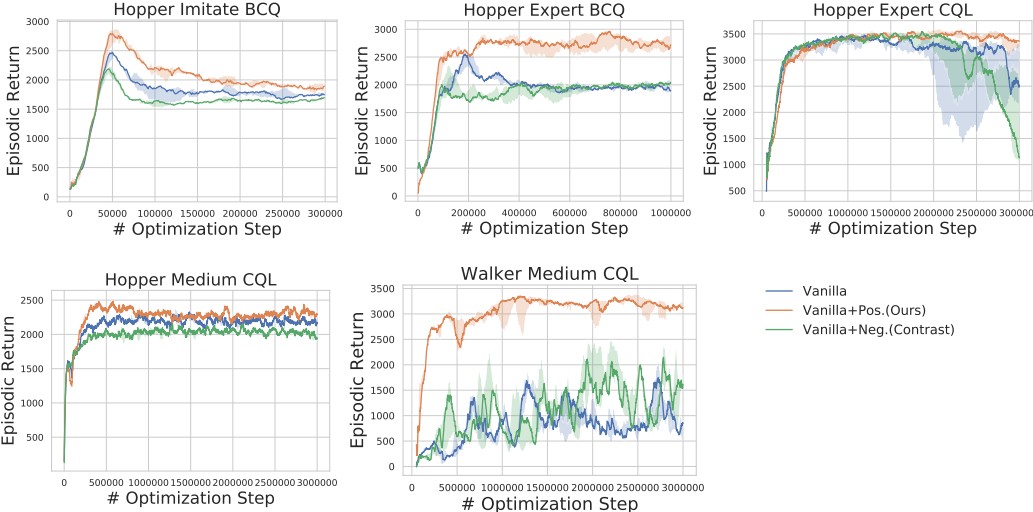

Figure 4: Results on offline RL settings. We verify our key insight that a positive reward shift equals to conservative exploitation thus helps offline value estimation, while a negative reward shift leads to worse performance.

et al., 2018a) and CQL (Bharadhwaj et al., 2020), i.e., both distribution-matching approach and conservative value estimation approaches in offline RL.

To verify our insight, we experiment with both positive reward shift (**Pos.**) and negative reward shift (**Neg.**), added on either BCQ or CQL. Figure 4 shows our experiment results. We experiment with both the dataset generated in BCQ (Hopper Imitate BCQ) and the dataset used in CQL (Fu et al., 2020) (others), and find in our experiments that learning with the CQL dataset is much more stable. the first two figures show results with BCQ as the backbone algorithm, where the former shows results on the vanilla BCQ dataset, while the latter shows results on the CQL dataset. The following three experiments in Figure 4 use CQL as the backbone. In all experiments, using a positive reward shift leads to improved learning performance, while a negative reward shift leads to performance decay, as expected. Implementation details and ablation study can be found in Appendix C.1.

## 5.2 ONLINE REINFORCEMENT LEARNING WITH RANDOMIZED PRIORS

We then conduct experiments on the online RL settings. We demonstrate our proposed method in the MuJoCo locomotion benchmarks. As our implementation is based on TD3, we use TD3-based variants as our baselines: The **TD3** is trained with default settings according to Fujimoto et al. (2018b). We also include **Ensemble TD3** and **Bootstrapped TD3** as baselines due to they are similar to our work in using multiple $Q$-networks in value estimation. We follow Osband et al. (2016) but extend it to the continuous control settings. Noting that in the continuous control setting, the argmax operator is approximated by the policy network, multiple policy networks are needed to cooperate with the multiple bootstrapped $Q$-value networks. Otherwise, multiple $Q$-value networks are not independent of each other thus breaking the condition of bootstrapped value estimation. The Ensemble TD3 presents the baseline performance when multiple $Q$-networks are used for value estimation in TD3, which also works as an ablation of our method when all reward shift priors are set to be $0$. As has been illustrated in Sec. 4.3.1, learning with different reward shifting values is equivalent to learning with optimistic or conservative initialization. In our method of Random Reward Shift (RRS), we use $3$ $Q$-networks with different priors. In our experiments, we found $\pm 0.5, 0$ works universally as a default setting. Though, further investigation on hyper-parameter may help to further improve the performance.

Results are shown in Figure 5. RRS outperforms the vanilla TD3 in all five environments and outperforms all baseline methods in most tasks. In all experiments, we use $3$ $Q$-networks for a fair comparison. Note that there is a trade-off between computational complexity and sample efficiency, i.e., using more $Q$-networks may further improve the performance at the cost of more computational

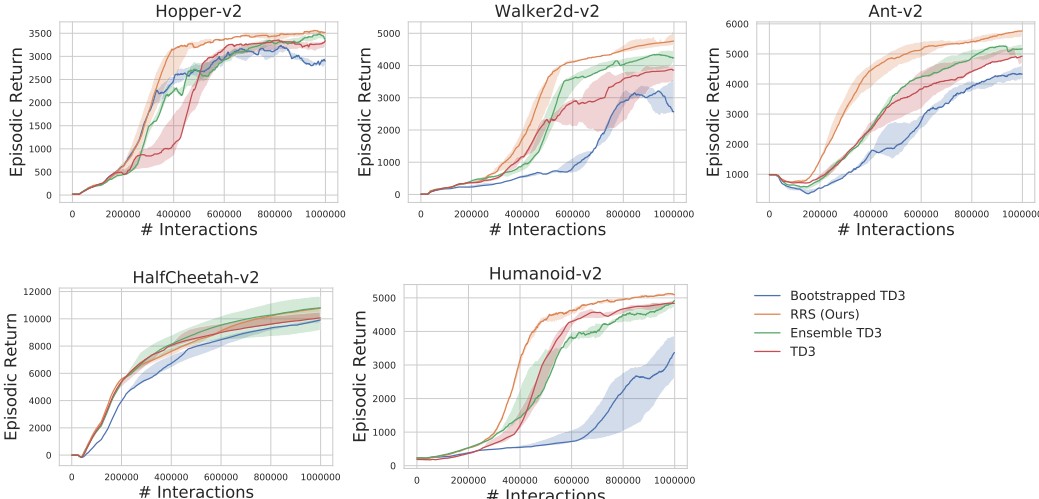

Figure 5: Results on continuous control tasks, the method of Random Reward Shift (RRS) outperforms its value-based baselines in most environments.

expensive, as reported in (Osband et al., 2016). More implementation details, pseudo code of RRS, and ablation studies can be bound in Appendix C.2.

### 5.3 Optimistic Random Network Distillation

Finally, we experiment with five discrete exploration tasks, namely the MountainCar-v0, and four navigation tasks of MiniGrid suite (Chevalier-Boisvert et al., 2018), namely the task of Empty-Random, MultiRoom, and FourRooms, to verify our insight on improving RND for value-based curiosity-driven exploration. More environment details are provided in the Appendix C.3.

We compare the vanilla **DQN**, vanilla **RND**, as well as improved DQN and RND according to our proposed insight. **DQN -0.5** indicates the results with a reward shifting of $-0.5$. Comparing DQN -0.5 with the vanilla DQN, our insight of negative reward shift leads to curiosity-driven exploration is again verified. **RND -1.0** indicates the results when a reward shifting of $-1.0$ is added to RND based on DQN. RND -1.0 improves the performance of RND and DQN in most environments, showing the effectiveness of the RND-based exploration bonus. Note that for a fair comparison, RND -1.0 should be compared with the vanilla DQN as the $-1.0$ reward shift just cancels the positive exploration bonus introduced by RND.

Moreover, we can further improve RND by equipping it with reward-shifting-based curiosity-driven exploration. RND -1.5 (i.e., RND with a $-1.5$ reward shift) can be compared with the DQN -0.5, as both receive an -0.5 exploration bonus for unseen states. We find in all experiments that our negative reward shift can remarkably improve exploration, not only working with RND to improve its performance but also work effectively in isolation.

## 6 Conclusion

In this work, we study how reward shifting affects policy learning in value-based deep reinforcement learning algorithms. Although constant reward shifting should not change the optimal policy induced by the optimal value function, in practice such a constant shift does affect the function approximation. Our detailed analysis manifests the fact that a constant reward shift is equivalent to using different initialization in the value function approximation. Specifically, we show that a negative reward shift leads to curiosity-driven exploration, while a positive reward shift helps conservative exploitation. The proposed idea is then verified through a variety of application scenarios, including offline RL, sample-efficient continuous control, and curiosity-driven exploration in value-based methods. Thus reward shifting is a simple yet effective technique that deserves further investigation.

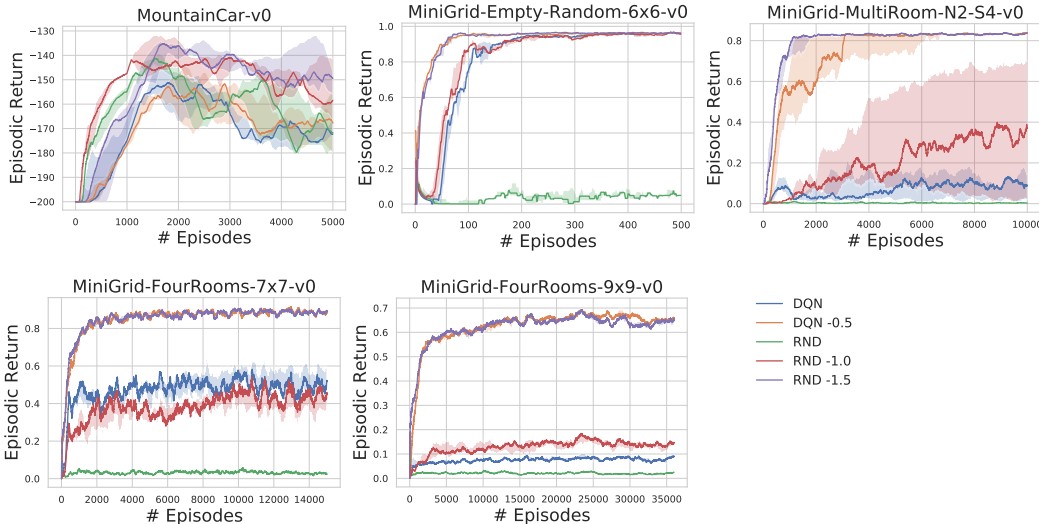

Figure 6: Value-based RND with shifted prior: Plugging the vanilla RND into DQN is not well-motivated according to our analysis in Section 4.3.2. The insight of equivalence between negative reward shifting and curiosity-driven exploration motivates us to shift the vanilla RND with a constant, which drastically improves the performance of RND when working with DQN.

## ETHICS STATEMENT

In this work, we study how the linear reward shaping in reinforcement learning benefits offline conservative estimation, online continuous control, and curiosity-driven exploration in discrete control tasks. Although in our work we experimented on a variety of benchmark environments, there are plenty of real-world applications: Improving the learning stability as well as the asymptotic performance of offline RL empowers the application of RL in scenarios where interaction with the environment is extremely expensive or unethical, e.g., healthcare, robotics, finance, etc. Moreover, our discussions on continuous and discrete control tasks open up a promising direction in pursuance of sample-efficient learning without introducing heavy extra computational burdens. Large-scale applications of our method can help improving efficiency in exploration, which is always conducted through computing additional curiosity networks explicitly in previous works.

## REPRODUCIBILITY STATEMENT

We include our code in the supplementary materials. More details for our experiments on offline RL, continuous control and discrete control can be found in Appendix C.1, Appendix C.2, and Appendix C.3, separately.

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

## A    PROOF OF PROPOSITION 1

*Proof.* the estimated $Q$-value $\hat{Q}(s,a)$ is composed by the two estimators with function approximation error, defined as $\epsilon_{b+}(s,a) = \tilde{Q}_{b+,t}(s,a) - \frac{b^+}{1-\gamma} - Q^*(s,a)$, and $\epsilon_{b-}(s,a) = \tilde{Q}_{b-,t}(s,a) - \frac{b^-}{1-\gamma} - Q^*(s,a)$.

$$
\begin{aligned}
&(1-\beta)\epsilon_{A,t} + \beta\epsilon_{B,t} \\
&= 2\eta(1-\beta)\hat{Q}^* + (1-\beta)(1-2\eta)\tilde{Q}_{A,t} + 2\eta\beta\hat{Q}^* + \beta(1-2\eta)\tilde{Q}_{B,t} \\
&= 2\eta\hat{Q}^* + (1-2\eta)[(1-\beta)\tilde{Q}_{A,t} + \beta\tilde{Q}_{B,t}] \\
&= 2\eta\hat{Q}^* + (1-2\eta)[(1-\beta)(1-2\eta)^t\tilde{Q}_{A,0} + \frac{4(1-\beta)\eta^2}{1-(1-2\eta)^t}\hat{Q}^* + \beta(1-2\eta)^t\tilde{Q}_{B,0} + \frac{4\beta\eta^2}{1-(1-2\eta)^t}\hat{Q}^*] \\
&= 2\eta\hat{Q}^* + (1-2\eta)[(1-2\eta)^t((1-\beta)\tilde{Q}_{A,0} + \beta\tilde{Q}_{B,0}) + \frac{4\eta^2}{1-(1-2\eta)^t}\hat{Q}^*] \\
&= \epsilon_{C,t}
\end{aligned}
$$

$$(7)$$

where $C = (1 - \beta)A + \beta B$ and the last line requires $\tilde{Q}_{A,0} = \tilde{Q}_{B,0} = \tilde{Q}_{C,0}$ are identical initialization.

With this notion, Equation (6) can be re-written as

$$
\begin{aligned}
\hat{Q}(s,a) &= Q^*(s,a) + (1 - \beta)\epsilon_{b^+}(s,a) + \beta\epsilon_{b^-}(s,a) \\
&= Q^*(s,a) + \epsilon_{(1-\beta)b^+ + \beta b^-}(s,a) \\
&= Q^*(s,a) + \epsilon_{c_r}(s,a)
\end{aligned} \tag{8}
$$

where the second line relies on the linear assumption of the approximation error $(1 - \beta)\epsilon_{b^+}(s,a) + \beta\epsilon_{b^-}(s,a)$. We further have $(1 - \beta)\tilde{Q}_{b^+} + \beta\tilde{Q}_{b^-} = \tilde{Q}_{(1-\beta)b^+ + \beta b^-}$ and $\hat{Q}(s,a) = \tilde{Q}_{c_r}(s,a)$, telling us that trading-off between the constant $b^-$ used for exploration and the constant $b^+$ used for exploitation with the coefficient $\beta$ is equivalent to use another constant with value of $c_r = (1 - \beta)b^+ + \beta b^-$.

$\square$

## B    IMPLICATIONS OF ASSUMPTION IN SEC. 4.3.1

In our main text, the estimated values for extremely o.o.d. samples are assumed to be near zeros. We provide detailed implications and explanations in this section.

On the one hand, it's clear that such an assumption holds for the tabular settings, that un-visited state-action pairs have the value in tabular initialization.

On the other hand, we acknowledge it as a mild assumption that there always exists o.o.d. samples that have the Q-values near zero for function approximation settings. Interpolation between those o.o.d. samples and other state-action pairs will clearly lead to an "in-between" value estimation, which in practice can be achieved with properly regularized neural networks.

The key insight we want to emphasize in Sec. 4.3.1 is that for frequently visited state-action pairs, the value discrepancy with different initialization are small, while for seldomly-visited state-action pairs, the discrepancy are relatively large, enabling the usage of such discrepancy as exploration bonus.

## C    IMPLEMENTATION DETAILS AND ABLATION STUDIES

**Hardware and Training Time**    We experiment on a server with 8 TITAN X GPUs and 32 Intel(R) E5-2640 CPUs. In general, shifting the reward does not introduce further computation burden except in the continuous control tasks, our method of Random Reward Shift (RRS) requires two additional $Q$-value networks. In our PyTorch-based implementation, those additional networks can be easily implemented and optimized in a parallel manner, and the extra computational burden is equivalent to using a $\sqrt{3}$ times wider neural network during optimization. It is worth noting that RRS is computationally much cheaper than the Bootstrapped TD3, where additional policy networks are also needed.

**Network Structure**    Our implementation of TD3, BCQ and CQL are based on code released by the authors, without changing hyper-parameters. We implement DQN based on a 3-layer fully connected neural network with 64 hidden units for the $Q$-value function, using ReLU and linear activation respectively. We use the Adam optimizer with learning rate of 0.001, and use an epsilon-greedy approach as naive exploration strategy. In our RND, we use two 4-layer fully connected neural networks with 512 units and ReLU activation in each hidden layer, and a softmax activation for the output layer. Adam optimizer is used for the optimization of the RND networks with learning rate 0.0001.

Our code is provided in the supplementary materials, and will be made public available.

## C.1 OFFLINE RL

In our experiments, we use a fixed dataset with $10k$ offline trainsition tuples for offline RL learning. Our implementation of BCQ and CQL are both based on the code provided by the authors. The only change we made to verify our insight is to shift the reward by a constant. In most environments, we find $r' = r + 8$ provides good enough performance. While in Hopper Medium CQL we find using a smaller positive reward shift $r' = r + 1$ works better than $r' = r + 8$, and for Walker Medium CQL, using a larger reward shift of $r' = r + 50$ further improves the result with $r' = r + 8$.

Figure 7 shows different performance under different choices of the reward shift constant. We denote a positive reward shift $r' = r + 8$ as **Pos.1**, denote $r' = r + 20$ as **Pos.2** and denote $r' = r + 50$ as **Pos.3** for all experiments excetp in the Hopper Medium CQL we use **Pos.1** to denote $r' = r + 1$.

In the experiments based on BCQ (first two figures). We can observe a uniformly performance improvement with all choices of reward shift constants. As the algorithm of CQL has already taken the conservative value estimation into consideration, in the experiments based on CQL, the performance is more closely related to the constant we use. Specifically, in Hopper Expert, while using any of the positive reward shift constants improve the learning stability, $r' = r + 8$ performs better on preserving the learning efficiency during early learning stage. For Hopper Medium, we find using larger positive constants hinder the performance. For Walker Medium, using a larger constant in reward shift performs much better than using a smaller one.

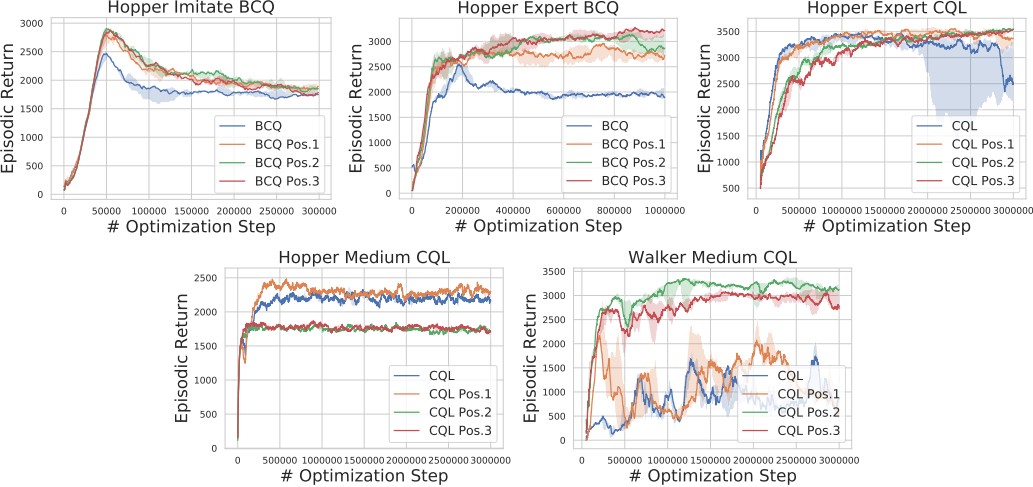

Figure 7: Performance with different reward shift constants.

## C.2 CONTINUOUS CONTROL

**Pseudo-Code for Random Reward Shift**   The pseudo-code of RRS is provided in Algorithm 1.

**Details of RRS**   Although we find in the motivating example that a $-5$ reward shift is able to remarkably improve the asymptotic performance of TD3, in this work we aim at proposing an uniformly suitable method based on the insight behind the motivating example. Therefore we propose to use $\pm 0.5, 0$ as the reward shifting constants. We find in experiment that the sampling frequency does not affect the performance. And in the experiments we follow BDQN Osband et al. (2016) to use a fixed value network throughout a whole trajectory. i.e., one of the $K$ $Q$-networks is sampled uniformly after each episode with length of 1000 timesteps. Intuitively, searching for more suitable reward randomization designs may further improve the performance, yet that is beyond the coverage of this work.

**Ablation Studies**   We experiment with different number of $Q$-value networks as well as different choices of the random reward shifting ranges. Results are presented in Figure 8. We denote RRS with 7 reward shifting constants ( and therefore also 7 $Q$-networks) as **RRS-**

---

**Algorithm 1** Sample-Efficient Continuous Control with Random Reward Shift

---

**Require**
- the size of mini-batch $N$, smoothing factor $\tau > 0$, $K$ reward shift values $r'_k = r + b_k, k = 1, \ldots, K$.
- Random initialized policy network $\pi_\theta$, target policy network $\pi_{\theta'}, \theta' \leftarrow \theta$.
- $K$ random initialized $Q$ networks, and corresponding target networks, parameterized by $w_k, w'_k, w'_k \leftarrow w_k$ for $k = 1, \ldots, K$. (e.g., a ModuleList in PyTorch).

**for** iteration $= 1, 2, \ldots$ **do**
    Uniformly sample one of the $K$ $Q$-functions, $Q_{w_k}$, for policy update
    **for** t $= 1, 2, \ldots$ **do**
        # Interaction
        Run policy $\pi_\theta$, and collect transition tuples $(s_t, a_t, s'_t, r_t)$.
        Sample a mini-batch of transition tuples $\{(s, a, s', r)_i\}_{i=1}^N$.
        # Update $Q_w$ (in parallel)
        Calculate the $k$-th target $Q$ value $y_{k,i} = r_i + b_k + Q_{w'_k}(s'_i, \pi_{\theta'}(s'_i))$
        Update $w_k$ with loss $\sum_{i=1}^N (y_{k,i} - Q_{w_k}(s_i, a_i))^2$.
        # Update $\pi_\theta$
        Update policy $\pi_\theta$ with $Q_{w_k}$
    **end for**
    # Update target networks
    $\theta' \leftarrow \tau\theta + (1 - \tau)\theta'$.
    $w'_k \leftarrow \tau w_k + (1 - \tau)w'_k, k = 1, \ldots, K$.
**end for**

---

**7**, and denote RRS with 3 reward shifting constants ( and therefore also 3 $Q$-networks) as **RRS-3**. The constants following **RRS-3/RRS-7** are the ranges of those random constants. Specifically, we use $[-0.5, 0, 0.5]$ for the **RRS-3 0.5** settings, $[-1.0, 0, 1.0]$ for the **RRS-3 1.0** settings, $[-0.5, -0.33, -0.17, 0, 0.17, 0.33, 0.5]$ for the **RRS-7 0.5** settings and $[-1.0, -0.67, -0.33, 0, 0.33, 0.67, 1.0]$ for the **RRS-7 1.0** settings. According to the experimental results, RRS is not sensitive to hyper-parameters, showing the robustness of the proposed method. We believe further search for those hyper-parameters can further improve the learning efficiency, yet this is off the main scope of this work and therefore left for the future research.

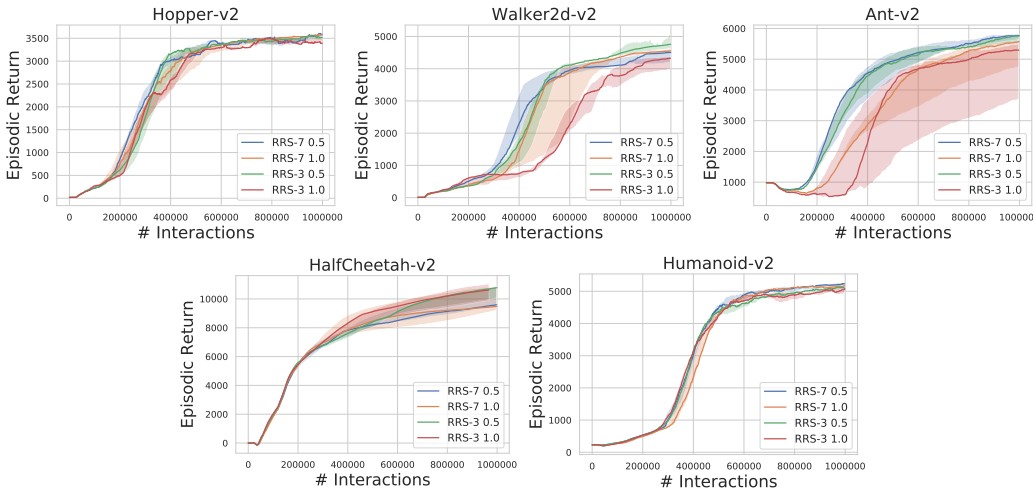

Figure 8: Performance with different reward shift constants and different number of $Q$-networks.

## C.3 RANDOM NETWORK DISTILLATION

**Environments** In this work, we experiment with five discrete (sparse reward) exploration tasks , namely the MountainCar-v0, and four navigation tasks of MiniGrid suite (Chevalier-Boisvert et al., 2018), namely the task of Empty-Random, MultiRoom, and FourRooms, to verify our insight on

improving RND for value-based curiosity-driven exploration. Figure 9 shows example of different tasks.

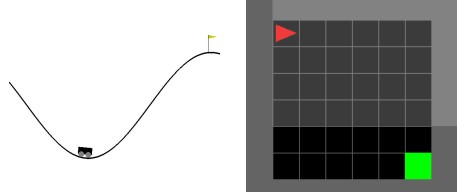 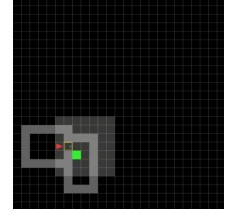 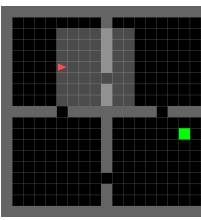

Figure 9: Examples of environments used in Section 5.3. The first figure shows the MountainCar-v0 environment where a car needs to accumulate potential energy to reach the flag, to receive a positive reward. The second figure shows the maze of the Empty-Random task with size of 6, the third one shows the MultiRoom of level S2-N4, where there are 2 rooms with size 4, the last figure shows example of FourRoom task with size 17. In our experiments, as we use the vanilla DQN as the baseline, which is not suitable for partial observable tasks, we use a smaller maze of size 7 and 9 to avoid further dependency on memories. In all tasks of the MiniGrid domain, the triangular red agent need to navigate to the green goal square, and the observable region is only a 7x7 square the agent is facing to (i.e., the regions with shallower color in the last three figures).

**Ablation Studies** We experiment with different reward shifting constants in the discrete control settings. We use a relatively large range in choosing constants, i.e., $\{-0.05, -0.15, -1.0, -1.5, -2.0, -2.5, -5.0, -10.0\}$. Results are presented in Figure 10. In all experiments, using a moderate reward shifting constant like $\{-1.0, -1.5, -2.0, -2.5\}$ remarkably improves the learning efficiency. On the other hand, a too aggressive reward shifting will lead to too much curiosity exploration and hinder the learning efficiency in the limited number of interactions.

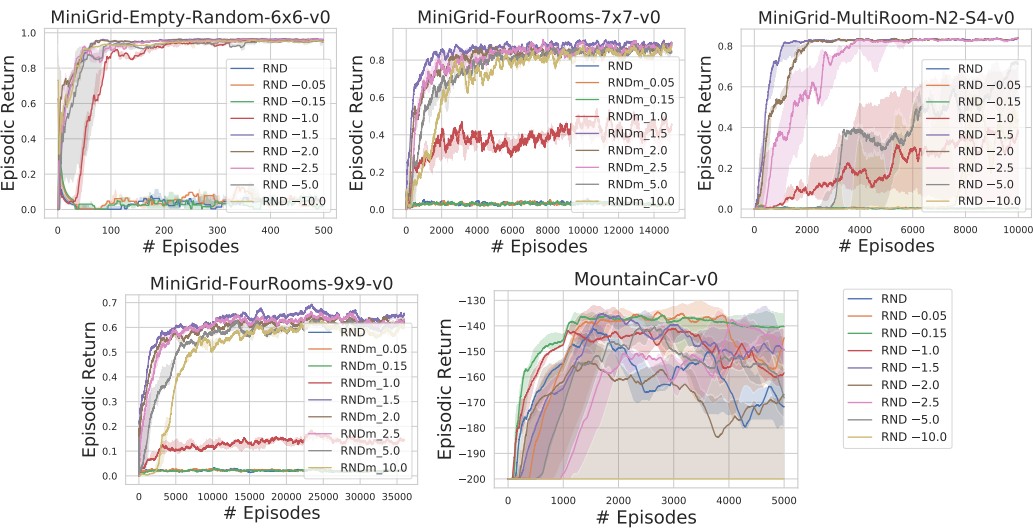

Figure 10: Performance with different reward shift constants in RND.

