# OpenReview forum: "Reward Shifting for Optimistic Exploration and Conservative Exploitation"
_ICLR.cc/2022/Conference — ICLR 2022 Submitted_

### Official Review · Reviewer_fFaW · 2021-10-31

**Correctness:** 3
**Technical Novelty And Significance:** 2
**Empirical Novelty And Significance:** 2
**Recommendation:** 3
**Confidence:** 4

**Main Review:**

Strength:
The strengths are summarized below:

1. This paper tries to unify optimistic exploration and conservative Q learning through reward shaping, the idea by itself is very interesting.
2. The paper provides several nice plots to help understand the algorithm.
3. The experiments are pretty solid including various settings (e.g., offline/online, discrete/continuous) and environments.

Weakness:
I would like to raise several questions to the paper:

1. In Sec.3 the motivating example, the design of the alive bonus is not just to provide the curriculum. Such design may also have the practical effect of preventing the robotic to break itself. For example, falling down can cause the robotic to break some of its components. In addition, since the reward is shifted, I am not sure how to read the curve in the figure. Is the orange curve representing the shifted reward and the blue one is the unshifted reward? If that is the case, they are not directly comparable. Thus, I think this moving example is problematic to some extent and needs more explanation.

2. In Figure. 2, it seems not correct that the initialization $\tilde{Q}_0$ is always smaller than $Q^*$. This would be interpreted as always having a pessimistic initialization of the $Q$ function.

3. In the same figure, the labeling: $|Q_t - Q^*| < |Q_{t-1} - Q^*|$ seems to be very strict and cannot be achieved in practice. Even we are doing optimization over the $Q$ function, I don't think it is a practical condition.

4. Same problem in Sec.4.2, the author states "the near-zero initialization guarantees the prediction is lower than the optimal Q-value, thus
conservative exploitation can be conducted with such a value function". Which I think is not valid, a simple negative reward case would cause this 0 initialization to be optimistic rather than conservative.

5. In the optimistic/pessimistic in face of uncertainty criterion, the algorithm design is always state-action dependent. Thus, I am having trouble understanding just adding a uniform bias can achieve such criterion, since it is not a function of state-action at all.

6. In the experiments, I would particularly like to see how the authors choose the value of bias. My feeling is that it will be very sensitive to the magnitude, but such magnitude is environment-dependent. I think adding an ablation on a larger range of values will help clarify this.

7. In addition, some of the other optimistic initialization/exploration designs are not sufficiently discussed. Such as https://arxiv.org/pdf/2002.12174.pdf.

Clarity: The paper is well-written and clear in the flow. In addition, the figure is pretty straightforward to illustrate the point.

Feedbacks & Questions: Please see details in the weakness.

**Summary Of The Paper:**

This paper shows a key observation that the linear transformation of reward shaping is equivalent to diversified Q-value network initialization. Based on such observation, the paper presents three scenarios where the reward shifting can benefit: the offline conservative exploitation, the online sample-efficient RL, and the curiosity-driven exploration. The algorithm is evaluated on the continuous control tasks and discrete control tasks.

**Summary Of The Review:**

Overall, I am still having some concerns about the claim of the paper. These concerns are pretty important and thus I encourage the author to engage in the discussion period and clarify these if there is any misunderstanding. I am happy to re-evaluate if the author convinces me. Based on my current evaluation, I don't recommend acceptance.

---

> ### Author Response · Authors · 2021-11-16
> **Response to Reviewer fFaW**
>
> Thank you for your thoughtful comments and suggestions. We give answers to each in turn.
>
> -----
>
>
> ### A1.
>
> Yes, in robotics applications there are safety concerns and physical constraints that can be cast to the threshold setting and early termination design. However, those topics are off the scope of this work. We focus on the linear reward shift that affects the Q-value function learning, which does not rely on any additional assumption or prior knowledge of the task.
> For the concerns on the comparison, we use different reward functions (i.e., the primal reward function with alive bonus, say $r$, and the revised reward function $r’ = r-5$) during training. The curves show the cumulative reward in evaluations, where both agents are evaluated on the primal task with $r$. So that they are comparable.
> We've revised the draft to better present the motivating curves and make it clearer accordingly.
>
> -----
>
>
> ### A2.
>
> Figure 2 is actually an illustrative figure that demonstrates the case when $Q_0$ is initialized universally smaller than the optimal value function $Q^*$. Later in our work, we can choose large enough $b^+$ as well as small enough $b^-$ to guarantee the optimal value function is bounded between (i.e., Figure 2(d) and Figure 3(b)).
> The $|Q_t - Q^*| < |Q_{t-1} - Q^*|$ also serves as an illustrative example to demonstrate the learning objective of temporal difference learning is to approximately learn the optimal value function, which induces the optimal policy.
>
> -----
>
>
> ### A3.
>
> Kindly let us reiterate the key insight for the offline-RL conservative exploitation to avoid potential misunderstanding. As we pointed in Figure 2, using a positive constant $b^+$ for reward shifting will increase the optimal Q-value $Q^*$ universally to $Q^*_{b^+} = Q^* + \frac{b^+}{1-\gamma}$. No matter what the primal reward value and the corresponding $Q^*$ are, we can find large enough $b^+$ such that $Q_0 < Q^*_{b^+}$.
> Or equivalently, we are initializing the Q-networks in our work with $\frac{b^+}{1-\gamma}$, rather than 0, for the primal optimal Q-value. We hope this can be clearer to address the reviewer’s concern.
>
>
> -----
>
> ### A4.
>
> Our design is inherently state-action dependent.
> Hark back to Figure 2, although we uniformly apply biases to all the states, this is not to say every state is visited equally frequently during learning. Thereafter, those under-explored state-action pairs will be affected more by the different initialization induced by reward shifting.
> A similar idea on inherent state-action dependence has been investigated in RND as well: the frequently visited state-action pairs will have a smaller discrepancy between the two networks’ outputs.
>
> -----
>
>
> ### A5.
>
> More ablation studies are provided in the updated draft for your reference.
>
> -----
>
>
> ### A6.
>
> We’ve updated the section of related work and included the discussion of the mentioned work.
>
>
> -----
>
>
> With our clarifications and updated presentations, we hope that we have addressed your concerns. Thank you for your kind consideration. Please let us know if you have further concerns and we are happy to address them.

---

### Official Review · Reviewer_4ENc · 2021-11-01

**Correctness:** 3
**Technical Novelty And Significance:** 2
**Empirical Novelty And Significance:** 2
**Recommendation:** 3
**Confidence:** 3

**Main Review:**

Strength:
- I do like the topic this paper studies, as I believe it is also what the deep RL community observes, that reward shifting does have an impact on the final learned policy.
- The paper discusses the reward shifting from both online and offline settings, which is interesting to see how positive/negative bias affects exploration and exploitation.
- Empirical studies in both online and offline setting verify the effectiveness of the reward shifting.

Weakness:
- The main argument in this paper is that reward shifting corresponds to different initialization, and pessimistic or optimistic initialization does affect the final learned policy, which is known to the community for a while. However, I think it lacks some intermediate important steps in the paper, such as how the initialization affects the learning dynamics? It seems all the arguments in the paper intuitively make sense, but lack strong theoretical support.
- The experiments though illustrates how the positive/negative bias helps in online/offline setting, which is well-known. A more important question is how to determine an appropriate bias in different tasks and environments. As shown in Appendix B.1, the magnitude of the bias does affect the learning behavior and outcome. It would be great to discuss this in a more principle way, instead of saying something like, $r+8$ seems work.
- Also, the author aims to study the linear transformation in the reward shift, but just ignores the scaling factor, which also seems have some significant effect in the learning process. It would be great to investigate more on this.

**Summary Of The Paper:**

This paper studies how different reward shifting (to be precise, adding a bias term on the reward) affects value-based RL algorithms. It illustrates the idea that different reward shifting corresponds to different initialization of the Q networks. Though linear reward shifting does not change the optimal solution for Q, the initialization does affect the learning process and might lead to different stationary points, when equipped with function approximation. The paper further studies the idea in two settings: (1). offline RL, which shows a position bias term could assign the $(s,a)\in\mathcal{D}$ a large value which helps conservative exploitation while in (2) online RL, a negative bias term could facilitate exploration. Some empirical results verify the ideas.

Reward shifting techniques have been studied in the bandit literature, this paper studies how it affects the value-based RL algorithms with function approximations. This is an interesting and important topic, and this paper provides some intuition and empirical results to help the community systematically understand the effects of reward shifting on both online and offline settings.

**Summary Of The Review:**

I like the topic this paper studies. However, the main argument in the paper is intuitive and seems well-known to the community. Beyond some empirical validation, I do not think it gives enough insight on how different initialization affects the learning dynamics and final learned policy. Also, there are important questions left, such as: (1). How to select the optimal bias $b$ to facilitate the learning process and (2). Does the scaling factor matter, empirically? Though the topic is interesting, given the current status of the paper, I would recommend a reject and I believe it would be a valuable paper if the authors study more deeply on the underlying causes of reward shifting.

---

> ### Author Response · Authors · 2021-11-16
> **Response to Reviewer 4ENc**
>
> Thank you for your time and efforts in the reviewing process, but **we can not agree with most of your comments.** We provide our responses in turn.
>
> ----
>
> ### A1.
>
> We admit our work is more on the side of empirical study rather than theoretical analysis. However, we believe the topic of our paper does conform to the ICLR submission guidance, where both empirical studies and theoretical studies are welcomed. In our experiments, we provide suggested hyper-parameters as other RL works do, and **we have never used the word ''seem’’, which is definitely inappropriate in academic discussions.** We tried to support our claim with empirical studies as well as analysis, though the latter is more on the heuristic side. We hope the reviewer can provide support for the comments.
>
> From the reviewers’ point of view, all practical algorithms should provide principled hyper-parameter tuning guidelines, it is totally off the central idea of our work. E.g., How can one principally determine how many units and how many layers to be used in our neural networks? We believe that, in general, neural networks are a good choice for a variety of tasks as one can easily find hyper-parameters that are suitable for a new given task. The basic idea of our recommended hyper-parameters serves the same idea: one is not expected to do heavy hyper-parameter tuning or search to apply our method to a new task.
>
> -----
>
> ### A2.
>
> We do not agree with the reviewer’s comment on  ''seems well-known to the community’’. To the best of our knowledge, we are the first to bring this idea into the broad DeepRL practice by demonstrating its effectiveness in offline and online settings. And we also do not think an important yet intuitive discovery can be anyhow weakened just due to it being intuitively correct and ''seems to be well-known’’.
> We ask the reviewer for further references to support the unfair claim.
>
> -----
>
>
> ### A3.
>
> **If the reviewer does read our paper thoroughly**, it is clear that we only discuss reward shifting, rather than reward scaling in this work as the latter is equivalent to adjusting the learning rate, which is clearly out of the scope of our paper. Adjusting the learning rate might be another interesting topic for the reviewer. Yet won't it be another ''well-known’’ topic that is ''not valuable’’?
>
>
> -----
>
>
> With our clarifications and updated presentations, we hope that we have addressed your concerns. Thank you for your kind consideration. Please let us know if you have further concerns and we are happy to address them.

---

### Official Review · Reviewer_Mgm9 · 2021-11-02

**Correctness:** 3
**Technical Novelty And Significance:** 3
**Empirical Novelty And Significance:** 3
**Recommendation:** 6
**Confidence:** 4

**Main Review:**

Strengths:
+ Overall, the paper is technically sound and the major claims are well supported with experimental results.
+ Various settings and scenarios are considered in the experimentation, including offline RL, online RL with continuous and discrete action spaces.
+ Related work in different domains, offline and online RL, is adequately cited.

Weaknesses/Concerns:
1.	[Clarity] Some parts of the paper are not clear enough in my opinion. Please find the list of items below for details.
-	1.a. It is not super clear to me how the Humanoid example in Section 3 connects to the major points made in this paper? Turning off the alive bonus is only applicable and effective for selective states, while the reward shifting is universal across all state-action pairs. To me, it only seems that the reward function for Humanoid is kind of problematic and is not designed well. Thus, it seems a bit off as a motivating example for reward shifting. Additional elaboration could be helpful here.
-	1.b. It is important to clarify the sign of parameter $b^-$ in the revision. Section 4.3 states that $b^-$ is a positive constant, while Figure 3 and Equation (6) should interpret $b^-$ as a negative constant I believe. Please make sure the definition of $b^-$ is consistent throughout the paper including Appendices.
-	1.c. It would be helpful to clearly state, for example at the end of the first paragraph under Section 4.3, that 4.3.1 is for continuous control and 4.3.2 is for improving curiosity-driven exploration with RND, so that the readers may be clearer that these two subsections are talking about different scenarios with their corresponding designs.
-	1.d. I had an impression that all the Q-functions should be seen as function approximators with neural networks. If that is the case, the statement from Section 4.3.1., “For under-explored state-action pairs, both $Q_A$ and $Q_B$ will give near-zero predictions as a consequence of initialization”, is not true to me. Even for unseen state-action pairs, the Q-network would extrapolate based on observed data, and the Q-values would not stay near-zeros for all under-explored state-action pairs. To be more rigorous, I encourage the authors to state this as an assumption in the paper or specify that this part of analysis only applies to tabular Q-learning settings.
-	1.e. The statement made in Proposition 1 is not clear enough. Appendix A definitely helps but I would still suggest elaborate a bit more in the main paper, especially for this part “combining the linear combination in Equation (6) is equivalent to using a linear combination of the constants”. Accordingly, it would be very helpful if the authors can describe more about the Random Reward Shift (RRS) in Section 4.3.1 or in Section 5.2. Again, I understand RRS much better by looking at Algorithm 1 in the Appendix, and I suggest make clearer connections between the algorithm RRS and Section 4.3.1, e.g., Equation (6) and how 4.3.1 leads to the design of RRS?
2.	[Experiments/Baselines]
-	2.a. How many seeds were used in the experiments Figure 4 - 6? I don’t think it is stated in the main paper?
-	2.b. To my understanding, RRS is training K = 3 different Q networks, each trained with reward shifting $b_k$, $k = 1, 2, 3$. Are these three shifting constants $b_1 = 0, b_2 = 0.5, b_3 = -0.5$ or they are sampled from $[0, 0.5, -0.5]$ randomly? In addition, it seems to me that RRS is similar to the ensemble of Q-networks with randomized additive priors [1]. Therefore, it is interesting to see how RRS would compare with [1] (or Ensemble TD3 + Randomized Prior Functions) when $K = 3$. It would be an interesting baseline to include in the paper.

Minor comments:
-	Remove the line break right below Figure 2.
-	Make sure the sign of $b^-$ is consistent in the paper.
-	Section 4.3.2, the penultimate paragraph, I believe it should be $I = max_{s, a} |A – B|^2$, the square is missing.

[1] Osband, Ian, John Aslanides, and Albin Cassirer. "Randomized prior functions for deep reinforcement learning." arXiv preprint arXiv:1806.03335 (2018).



**Summary Of The Paper:**

This paper studies the effectiveness of reward shifting in value-based deep reinforcement learning. Particularly, it points out that (1) a positive reward shifting is equivalent to pessimistic initialization of Q values, thus, leading to conservative exploitation; and (2) a negative reward shifting equals to optimistic initialization and hence leads to curiosity-driven exploration. By leveraging these insights, the paper proposes to modify existing RL algorithms by simply adding reward shifting to improve their performance. Empirically, experimental results showed that reward shifting helps improve the baselines (or the vanilla algorithms) under various scenarios, including offline RL, online continuous control and online value-based curiosity-driven exploration.

**Summary Of The Review:**

The paper provides helpful insights to better understand the effectiveness of reward shifting in deep reinforcement learning. Major claims are well supported by empirical experiments in various domains, including offline and online RL. My main concerns are about the clarity of the paper, which I believe needs to be further improved, and additional baselines (please see the above Concerns for details). Therefore, I vote for a weak reject for the current submission. I am willing to adjust my scores should my concerns be addressed in the rebuttal period.


=== Updated ===

My major concerns were addressed during the rebuttal period, so increased my score accordingly.

---

> ### Author Response · Authors · 2021-11-16
> **Response to Reviewer Mgm9**
>
> Thank you for your thoughtful comments and suggestions. We give answers to each in turn.
>
> -----
>
>
> ### Clarity
>
> -----
>
>
> - a. In fact the alive bonus is added to all effective states: as there is early termination, any state that is not assigned with the alive bonus will never be added to the replay buffer for Q-value and policy learning. Therefore, it is equivalent to regard the alive bonus as a universal reward shift.
> Our claim following this motivating example is, a positive reward shift leads to conservative behaviors thus hindering the learning of the agent.
>
>     We use Humanoid as the motivating example because it is well-recognized as the most challenging environment in the MuJoCo locomotion benchmarks. DRL algorithms can hardly learn a policy with more than 5k rewards, while a simple reward shifting helps to improve the score to more than 7k, which is really impressive.
>
> - b. We’ve corrected our notion for the negative bias constant $b^-$ in the updated draft.
>
> - c. We’ve updated the draft accordingly in Sec. 4.3 to make it clearer and better organized.
>
> - d. We’ve updated our presentation on this assumption accordingly.
>
>     On the one hand, it’s clear that such an assumption holds for the tabular settings, that un-visited state-action pairs have the value in tabular initialization.
>
>     On the other hand, we acknowledge it as a mild assumption that there always exists o.o.d. samples that have the Q-values near zero for function approximation settings. Interpolation between those o.o.d. samples and other state-action pairs will clearly lead to an ``in-between’’ value estimation, which in practice can be achieved with properly regularized neural networks.
> The key insight we want to emphasize in Sec. 4.3.1 is that for frequently visited state-action pairs, the value discrepancy with different initialization is small, while for seldomly-visited state-action pairs, the discrepancy is relatively large, enabling the usage of such discrepancy as exploration bonus.
>
> - e. We’ve updated our presentation on Proposition 1 and tried to make clearer connections between the algorithm RRS and Section 4.3.1. We elaborated the details of RRS in the appendix during submission due to the page limit.
>
> -----
>
>
> ### Experiments/Baselines
>
> -----
>
>
> - a. We experiment with $8$ random seeds for all experiments except in the motivating example, where we used $5$.
>
> - b. There are three shifting constants fixed for each Q-value network across learning. And for policy updates, we randomly use one of those three Q-networks to estimate the policy gradient. The updated draft better presents our implementation detail for RRS.
>
> - c. Although it sounds like a good idea to combine our method with the work of random prior DQN [1]. In fact, it is not suitable for continuous control tasks we consider in our work. There are several non-trivial problems to solve before adopting it into our setting: 1. Finding a suitable set of random prior functions is challenging, though our work can be regarded as a kind of (very) special case of having constants as those prior functions and 2. [1] uses neural networks as random priors, which is equivalent to fixing part of the value network during training. However, the problem is the ''argmax’’ operator in [1] is applied on top of $f+p$, meaning the prior function will have an impact on the optimal policy, although this effect can be averaged out with a sufficiently large amount of ensemble networks, we do not consider it to be practically tractable with the fact that 3. its computational expense is even higher than BDQN, as the algorithm requires multiple value networks and multiple policy networks.
>
>     We do not want to adjust the primal algorithm with arbitrary choices as it can hardly be a fair comparison without an elaborated adaptive design. We are happy to discuss further with the reviewers on this related work and run some experiments once we converge to some concrete ideas for implementation.
>
>
> -----
>
> With our clarifications and updated presentations, we hope that we have addressed your concerns. Thank you for your kind consideration. Please let us know if you have further concerns and we are happy to address them.
>
>
> -----
>
>
> ### Reference
>
> [1] Osband, Ian, John Aslanides, and Albin Cassirer. "Randomized prior functions for deep reinforcement learning." arXiv preprint arXiv:1806.03335 (2018).

---

> > ### Comment · Reviewer_Mgm9 · 2021-11-20
> > **Thank you and my concerns are addressed**
> >
> > Thanks for the detailed response and the revision. My major concerns are resolved now. I believe the comments on the additional baseline are valid, and that adapting the randomized prior functions to continuous control may require some additional work and thoughts, which is not the main focus of this paper. However, it is still worth mentioning in the revision that “our work can be regarded as a kind of (very) special case of having constants as those prior functions” or “[1] can be seen as having randomized reward shifting injected in Q-function learning”. Also, RRS and [1] both reply on the “diversity” of value estimates from different Q networks to induce exploration. Please make it clear in the paper as well that with simple reward shifting, RRS is applicable and targeting continuous control while [1] is limited to discrete problems at the moment. In addition, I like the ablation studies in the updated paper, which is helpful.
> >
> > Overall, my concerns are addressed and I will adjust my score and update the review accordingly.
> >
> > [1] Osband, Ian, John Aslanides, and Albin Cassirer. "Randomized prior functions for deep reinforcement learning." arXiv preprint arXiv:1806.03335 (2018).

---

> > > ### Author Response · Authors · 2021-11-20
> > > **Thanks for your feedback!**
> > >
> > > -----
> > >
> > > Thank you very much for the insightful and positive comments!
> > >
> > > We've uploaded a new version of the paper which contains a more detailed discussion on the relationship and differences between [1] and reward shifting (Page 3, Sec. 2.2).
> > >
> > > We've also added the ablation study on the last environment in Figure 10, which takes a longer time than the others to finish. Our conclusions for the ablation studies get further verified based on the results.
> > >
> > > Thanks again for your time and efforts!
> > >
> > >
> > > -----
> > >
> > > [1] Osband, Ian, John Aslanides, and Albin Cassirer. "Randomized prior functions for deep reinforcement learning." arXiv preprint arXiv:1806.03335 (2018).

---

### Official Review · Reviewer_dgHr · 2021-11-03

**Correctness:** 4
**Technical Novelty And Significance:** 2
**Empirical Novelty And Significance:** 2
**Recommendation:** 5
**Confidence:** 3

**Main Review:**

Advantage:
With the idea of reward shifting, this paper considered different reward shifting schemes for a lot of different scenarios such as offline RL, online RL, and curious-driven RL. The intuition of using pessimistic Q-value for offline RL and optimistic Q-value to encourage exploration is easy to understand.

Some questions:
1. Is constant reward shifting can be regarded as solving a different MDP or is actually a way to design reward but not a more efficient way to accelerate the Q-value convergence?
2. It seems to minus/plus a constant reward to the Q-estimates can be seen as using pessimistic penalty in offline RL or optimistic bonus for exploration in online RL. The previous works designed different values of the bonus $b(s, a)$ corresponding to different $(s,a)$ pairs, which may be more efficient than adding a constant reward for all the Q-values? However such kinds of baselines seem not to be covered or even discussed? I'm not sure whether they are related such as [1],[2].
3. It is claimed in Section 2.1 that this paper considers off-policy cases? So will the data be generated under a fixed behavior policy or generated conditioned on some Q-values?
4. It is a little bit counterintuitive why changing the initialization of Q-value will bring benefits for the algorithms. Maybe the biggest change brought by reward shifting is to change the landscape of the Q-value (or say the optimal Q-values after changing the reward) but not the change of the initialization? If that landscape and all the local and global minima of a loss function does not change, changing the initialization helps just means the algorithm is not robust which depends a lot on the initialization?
5. Will the initialization of Q-values be too conservative or optimistic which may hinder the performance of the algorithm? And how to choose the reward shifting magnitude?
6. It seems a lot more baselines involving other kinds of pessimistic or optimistic additional terms instead of constant reward shifting will be more convincing?

[1] Jin, Chi, et al. "Is Q-learning provably efficient?." arXiv preprint arXiv:1807.03765 (2018).

[2] Rashidinejad, Paria, et al. "Bridging offline reinforcement learning and imitation learning: A tale of pessimism." arXiv preprint arXiv:2103.12021 (2021).

**Summary Of The Paper:**

This paper proposes a linear combination of reward shifting to improve the performance of value-based reinforcement learning. Using the equivalence between reward shifting and a new initialization for the Q-value function approximation, this work shows that the optimal policy will not change and the method can achieve better results compared to no reward shifting.

**Summary Of The Review:**

In my opinion, this work is slightly below the acceptance threshold. But I'm happy to change my score if the problems are addressed well.

Advantage:
It formulates the addition of a pessimistic penalty or an optimistic bonus when learning Q-value estimations as a reward shifting, which is may of independent interest.
Targeting different settings in RL, it designs different algorithms using reward shifting.

Disadvantage:
I'm not sure whether the intuition that changing the initialization of the Q-value network is the main reason why the algorithm with reward shifting outperforms other baselines.
If we see reward shifting as an extra term added for Q-values, there are some other ways instead of adding constant to do that, while there are no baselines about that is involved.

---

> ### Author Response · Authors · 2021-11-16
> **Response to Reviewer dgHr**
>
> Thank you for your thoughtful comments and suggestions. We give answers to each in turn.
>
> ---------
>
> ### A1.
>
> As constant reward shifting does not change the optimal Q-function as well as the corresponding optimal policy induced by Q-function, we are indeed solving the save MDP. Although in tabular settings or bandit settings we are able to provide convergence analysis of a given algorithm, in deep reinforcement learning settings such analysis is always intractable.
> In this work, we demonstrated that with a limited number of samples (rather than infinite many interactions in theoretical analysis), reward shifting is helpful in improving learning efficiency for online settings and improving learned policies for offline settings. Noted that for the offline settings, the convergence values of Q-value with different initialization are clearly different. We claim in our work with a conservative reward shifting (i.e., with a positive bias), the converging Q-value is much more reliable in terms of evaluating-time extrapolation.
>
> ---------
>
> ### A2.
>
> We will discuss those (theoretical-side) related works [1], [2] in our revision. Yet kindly let us reiterate the difficulty we met in Deep-RL practice is that we can not directly calculate those provably-efficient curiosity bonus $b(s,a)$’s. Instead, in our work, we discussed two heuristic variants of those count-based curiosity approaches in Sec. 4.3 [3], [4].
>
> ---------
>
> ### A3.
>
> For the online-RL settings, our experiments are built upon off-policy RL algorithms (TD3, DQN), and for the offline-RL settings, our experiments are based on CQL and BCQ, which are by definition off-policy as in offline-RL settings there are no more interactions with the environment permitted.
> By definition, off-policy does not mean we have a fixed-Q value during learning, but we can re-use previous samples generated by earlier (probably poorer-performing) policies in pursuance of higher sample efficiency. Off-policy RL algorithms are in general much more sample-efficient than on-policy ones in practice and are thus more promising in real-world applications.
>
> ---------
>
> ### A4.
>
> ''the biggest change brought by reward shifting is to change the landscape of the Q-value’’ Yes exactly. The optimal Q-values are uniformly changed, which does not affect the optimal policy that only considers the optimal action with regard to the Q-values (by an ''argmax'' operator, either precisely argmax or policy network in actor-critic algorithms.).
> Noted that our claim is that different initializations lead to different exploration behaviors during learning due to the argmax operators being applied to the imperfect Q-values during learning. We believe it is worth reiterating that a uniform negative reward-shifting is equivalent
> Moreover, it might be useful to reiterate that in DeepRL practice it’s in general hard to claim a policy has learned the optimal policy, and considering even in supervised learning, initialization matters [5]. It is totally acceptable for DeepRL, where the i.i.d. assumption on the dataset no longer holds, also suffers from suboptimal behaviors due to bad initialization.
>
> ---------
>
> ### A5.
>
> In general, adjusting such shifting terms for different tasks may gain more performance improvement, as different domains have a different specification of reward scaling, etc, this is not a hyper-parameter of the algorithm but also related to the task domains.
> However, in our work, we tried to demonstrate the proposed idea by emphasizing the universal effectiveness of such reward shifting in both conservative exploitation and optimistic exploration. In our work, the reported main results with universal hyper-parameters. And ablation studies are detailed in the appendix.
> We admit that investigating deeper on the design of such shifting bias is important and interesting, however, they are not included in the current scope of this paper. Think about the fact that people never require RL algorithms (e.g., PPO/SAC/TD3, etc.) to perform universally good for all domains with a single set of hyper-parameters.
>
> ---------
>
> ### A6.
>
> The basic idea of our experiments is to provide comparisons: as our proposed method is orthogonal to previous methods with additional curiosity terms (or pessimism punishments), in principle our proposed method can be plugged into any algorithm with proper adaptation. In our work, we choose the most classic and prevailing (SOTA) algorithms to demonstrate the general effectiveness of the proposed method to lay stress on the idea itself and make our presentation clear.
>
> ---------
>
> With our clarifications and updated presentations, we hope that we have addressed your concerns. Thank you for your kind consideration. Please let us know if you have further concerns and we are happy to address them.

---

> > ### Author Response · Authors · 2021-11-16
> > **Response to Reviewer dgHr**
> >
> > ---------
> >
> > ### References
> >
> > [1] Jin, Chi, et al. "Is Q-learning provably efficient?." arXiv preprint arXiv:1807.03765 (2018).
> >
> > [2] Rashidinejad, Paria, et al. "Bridging offline reinforcement learning and imitation learning: A tale of pessimism." arXiv preprint arXiv:2103.12021 (2021).
> >
> > [3] Ian Osband, John Aslanides, and Albin Cassirer. Randomized prior functions for deep reinforcement learning. In Advances in Neural Information Processing Systems, pp. 8617–8629, 2018.
> >
> > [4] Yuri Burda, Harrison Edwards, Amos Storkey, and Oleg Klimov. Exploration by random network distillation. arXiv preprint arXiv:1810.12894, 2018b.
> >
> > [5] Daniely, Amit, Roy Frostig, and Yoram Singer. "Toward deeper understanding of neural networks: The power of initialization and a dual view on expressivity." Advances In Neural Information Processing Systems 29 (2016): 2253-2261.

---

> > ### Comment · Reviewer_dgHr · 2021-11-21
> > **Response**
> >
> > Thanks for the author's response. The practical findings are very interesting. I got all the questions solved. However, in my opinion, more insights and underlying reasons of why the changing of the initialization will help the optimization process are needed to convince readers, such as the analysis of the landscape of the Q-values. So, unfortunately, I will keep my score.

---

### Decision · Program_Chairs · 2022-01-20

**Decision:**

Reject

**Comment:**

I thank the authors for their submission and active participation in the discussions. The majority of reviewers have concers with this paper, in particular, regarding the motivation of the method [dgHr], clarity [Mgm9], and theorethical support [4ENc]. I side with reviewers 4ENc, dgHr and fFaW, and recommend rejection of this paper. I want to encourage the authors to use the feedback by the reviewers to improve their paper.